# Derailment risk: A systems analysis that identifies risks which could derail the sustainability transition

Laurie Laybourn[1], Joseph Evans[2], James Dyke[1]

[1] Global Systems Institute, University of Exeter, Exeter EX4 4QE, UK

[2] Institute for Public Policy Research (IPPR), London SW1P 3AY, UK

*Correspondence to*: Laurie Laybourn (l.laybourn-langton@exeter.ac.uk)

**Abstract.** The consequences of climate change, nature loss, and other changes to the Earth system will impact societies' ability to tackle the causes of these problems. There are extensive agendas of study and action on the risks resulting from changes in the Earth system. These consider the failure to realise rapid sustainability transitions to date ("physical risk") and the risks resulting from these transitions going forward ("transition risk"). Yet there is no established agenda on the risks *to* sustainability transitions *from* both physical and transition risks and their knock-on consequences. In response, we develop a conceptual socio-ecological systems model that explores how the escalating consequences of changes in the Earth system impacts the ability of societies to undertake work on environmental action that, in turn, re-stabilises natural systems. On one hand, these consequences can spur processes of political, economic, and social change that could accelerate the growth in work done, as societies respond constructively to tackle the causes of a less stable world. Conversely, escalating demands to manage increasingly chaotic conditions could divert work and political support from environmental action, deepening changes in the Earth system. If the latter dynamic dominates over the former, the chance is increased of passing a planetary threshold over which human agency to re-stabilise the natural world is severely impaired. We term this 'derailment risk': the risk that the journey to bring the world back into a safe operating space is derailed by interacting biophysical and socioeconomic factors. We use a case study of a climate tipping element - the collapse of the Atlantic Meridional Overturning Circulation (AMOC) - to illustrate derailment risk. A range of policy responses can identify and mitigate derailment risk, including transformational adaptation. Acting on derailment risk is a critical requirement for accelerating the re-stabilisation of Earth system elements and avoiding catastrophic outcomes.

## 1 Introduction

How will the effects of climate change, nature loss, and other environmental change impact our ability to tackle the causes of these problems? There is already a high demand on resources to respond to worsening climate shocks, knock-on impacts for areas such as food production and health, and the many other growing consequences of changes to the Earth system (Pörtner et al., 2022). These impacts are expected to increase in a warmer future, placing ever greater demands on our attention and resources as we respond to worsening

conditions and larger crises. Meanwhile, an increasingly turbulent world could impact our ability to coordinate responses to escalating crises and to address the underlying causes, including through disrupting international cooperation (Millward-Hopkins 2022). So it is important to explore how the growing demands of a world made more chaotic by the climate and ecological emergency could impact policy strategies intending to respond to that emergency.

Policymakers currently consider a range of risks resulting from climate change and other environmental destabilisation. For example, frameworks used by government agencies and central banks to explore the financial and economic risks resulting from climate change identify two main categories (FSOC 2021; TFCD 2021). Firstly, the 'physical risks' of climate change. These relate to the physical impacts on societies, such as rising temperatures eroding labour productivity. Secondly, the 'transition risks' resulting from action to reduce greenhouse gas emissions. These include the problem of 'stranded assets', such as the loss of investments in coal power plants that must be closed before their planned end of life as fossil fuel use is rapidly curtailed. Scenarios using these risk categories explore how a faster transition to net-zero emissions globally will reduce physical risks while increasing transition risks, and vice versa (NGFS 2022). Guided by this influential framework, policymakers aim to manage these risks, often quantifiable in terms of monetary costs, by optimising strategies that balance physical and transition risks. With assumptions of economic growth and technological advancement, a global solution to these risks seems attainable. This would minimise the excursion from Holocene conditions, and thus increase the chances for humanity to remain within a 'safe operating space' (Rockström et al., 2023).

However, there remains a dangerous gap when it comes to the assessment of risks to the transition itself. These risks emerge from the deepening consequences of changes in the Earth system, which might act as a drag on economic growth, deter global cooperation, and cause other effects that frustrate our collective ability to deliver rapid re-stabilisation of biophysical systems (Franzke et al., 2022). While a cost optimised transition might exist in theory, its implementation in practice could be slowed, even blown off course, by the impacts of climate and ecological change. This points to a third category of risk. Not just the risks from the failure to realise a rapid transition to date (physical risk), nor the risks from the transition going forward (transition risk), but the risk *to* the transition from both physical and transition risks and their knock-on consequences.

The risks to societies that arise because of a slower transition - resulting from increasing impacts from climate and ecological change - are typically considered as exogenous. These impacts are imposed on societies. The risks that arise because of the transition are considered as endogenous. These impacts are generated by the transition itself. To understand the risks *to* the transition consequently requires a complex interaction of exogenous and endogenous factors. We term this as 'derailment risk': the risk that humanity's efforts to remain with a safe operating space are derailed by interacting biophysical and socioeconomic factors.

To explore this novel concept, this article considers the interactions between derailment risk and existing concepts of physical risk and transition risk. To do so, we develop a conceptual model that explores the consequences of these interactions on socio-ecological systems. This necessarily requires analysis that involves multiple feedback loops. Our primary focus lies in identifying possible destabilising dynamics operating between biophysical and socioeconomic systems that could erode the ability of global society to accelerate (or

even maintain) the sustainability transition. These feedback loops could degrade attempts at emissions reductions, nature restoration, and other actions intended to reduce impacts and then re-stabilise biophysical systems. If the pace of the transition were to fall below critical values, then the risk of activating tipping elements in the Earth system would increase. Activating biophysical tipping points would increase the strength of reinforcing feedback loops, creating a catastrophic dynamic in which a cascade of feedback loops between accelerating Earth System destabilisation and socio-economic consequences becomes irrevocable. We use the example of the collapse of the Atlantic Meridional Overturning Circulation (AMOC) as an example of a climate tipping event that could present severe risks to global transition efforts if it were to occur. This is because of the severe impacts this event could have on the global food system and other systems critical to the security of societies and ecosystems (OECD, 2021).

Conversely, it is also possible that the destabilising dynamics operating between biophysical systems and human societies can create opportunities for acceleration of the transition via rapid socioeconomic change. For example, increased awareness of impending destabilising feedback loops and coordinated policy interventions could initiate the activation of 'positive' tipping points; these are reinforcing feedback loops operating in social, economic, and political systems that could produce a step-change in action to re-stabilise elements of the Earth system (Sharpe and Lenton, 2021; Winkelmann et al, 2022). This offers a possible pathway for effective stewardship of the Earth system into a safe space for humanity over the coming decades (Rockström et al., 2023). Anticipating and managing risks to the transition - derailment risk - is therefore of paramount importance. Both protecting and enhancing our collective ability to trigger positive tipping points in social systems that would accelerate sustainability action, even as chaotic events grow, could create a more powerful counter-effect, avoiding spiralling disaster. In developing a simple qualitative model, this article hopes to be of value to the research community, policy makers, and wider society in driving understanding and action on derailment risk.

## 2 Derailment risk

To conceptualise the process of transition to a safe space for humanity, we extend the physical concept of work within a socio-ecological systems model. In our broad conception, work includes labour and physical work done and the resource inputs for this work, including financial and material resources, as well as time and less tangible inputs, such as attention. The socio-ecological systems model is illustrated in figure 1.

By the end of the 20th century human energy use had reached a magnitude comparable to the biosphere (Lenton et al., 2016). Most of the energy provided to power this work has been provided by fossil fuels. The burning of coal, oil, and gas since the industrial revolution has released 1.5 trillion tons of carbon dioxide into the atmosphere, while to date humans have affected three quarters of the Earth's total ice-free land surface (Arneth et al 2019). The net result of this prodigious energy and material consumption is dangerous interference in the Earth system (Rockström et al., 2023). The planetary boundaries framework identifies climate change, biodiversity loss, and biogeochemical cycle disruption as the most at risk of nine Earth system functions (Steffen et al., 2015).

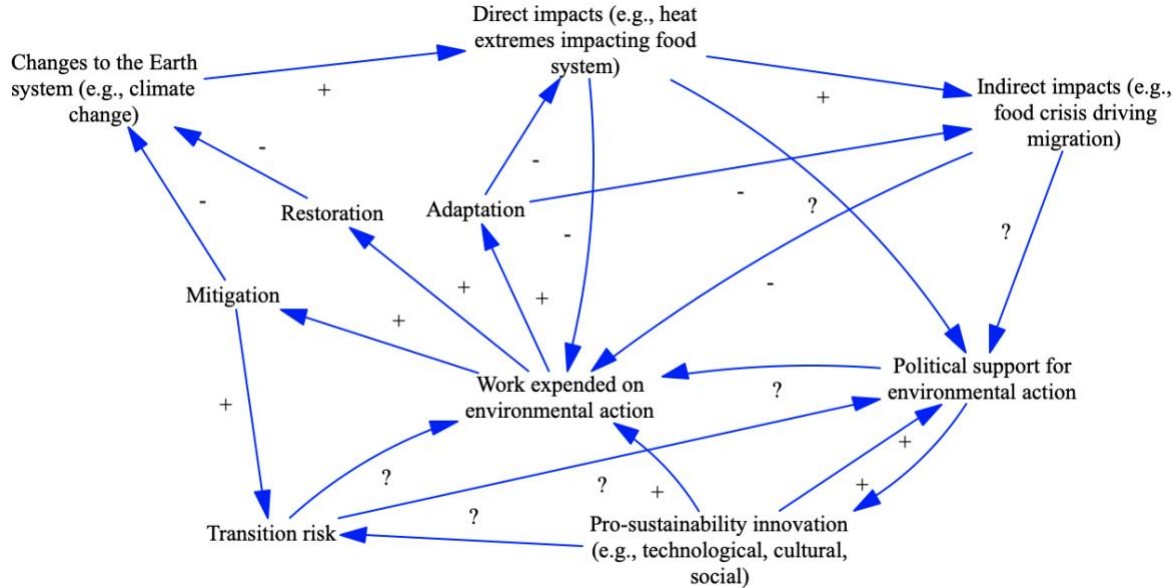

**Figure 1: Illustration of derailment risk using the feedback mechanisms between the work done by societies to re-stabilise elements of the Earth system and how these are, in turn, impacted by the direct and indirect impacts that result from changes in the Earth system. Positive polarities - where element A has an increasing effect on element B - are illustrated with a + sign. Negative polarities - where element A has a decreasing effect on element B - are illustrated with a - sign. Ambiguous polarities - where the overall effect is unclear - are illustrated with a question mark. Polarities are assigned through literature review.**

As figure 1 shows, we consider that societies seek to address changes in the Earth system and their consequences through undertaking three broad categories of work. Firstly, mitigation of environmental harms, including through rapidly reducing greenhouse gas emissions and halting biodiversity loss. Secondly, adaptation to the inevitable consequences of current and future destabilisation of Earth system elements. Thirdly, the restoration of human impacts on Earth system elements, such as through ecosystem restoration and carbon dioxide removal. Together, these three areas of work - mitigation, adaptation, and restoration - constitute the process of an overall sustainability transition whereby societies seek the progressive re-stabilisation of biophysical systems.

Restoration is often assumed within mitigation. For example, the United Nations Framework Convention on Climate Change (UNFCCC) recognises the role ecosystems such as forests have in sequestering carbon dioxide (UNFCCC, 2023). We have separated out mitigation and restoration in our analysis in recognition of the sheer scale of restoration needed to re-stabilise the different elements of the Earth system. For example, policy trajectories for climate action assume a large and increasing burden of carbon dioxide removal from the atmosphere, which will require younger and future generations to undertake significant work to meet (Hansen et al., 2017). This will need to be done at the same time as meeting all mitigation and adaptation requirements. There will be limits to the amount of work that current and future generations will be able to commit to this

effort. In figure 1, we identify a total amount of work available for mitigation, adaptation, and restoration: 'work expended on environmental action'. This is mediated in four ways.

**2.1 Interactions that affect the work expended on environmental action**

Firstly, we assume that the direct and indirect consequences of destabilisation of elements of the Earth system will decrease the amount of work available, as illustrated in figure 1 by the negative polarity between both direct and indirect effects and the work expended on environmental action. For example, direct impacts such as periods of extreme heat and humidity will reduce the amount of work available to be expended on environmental action by eroding labour productivity (Dasgupta et al., 2021). Indirect impacts - which

encompass the socio-economic consequences of environmental change - can also decrease the work available for environmental action. For example, prolonged periods of extreme heat can lead to food production losses (Zhao et al., 2017). In turn, the socio-economic impacts of food production losses can include increased poverty and migration, which cause knock-on economic disruption and political destabilisation (Chatham House, 2021). Such effects can be transmitted around the world through globalised socio-economic systems and lead to what

can be considered as maladaptive responses such as food export limitations in attempts for nations to reduce their exposure to food insecurity by hoarding. These destabilising dynamics interact with and exacerbate existing social, economic, and political challenges (Keys et al., 2019). In this way, we can see how the initial direct impacts of environmental change (here leading to food crises) can produce reinforcing feedback loops that serve to draw finite resources away from working directly on responding to the climate and ecological

crisis. By helping societies better cope with environmental shocks, adapting societies to direct and indirect impacts should lead to more resilience in societies' abilities to continue to work on the sustainability transition. This is illustrated in figure 1 by the negative polarity between adaptation and both direct and indirect effects.

Secondly, we assume that direct and indirect impacts of the destabilisation of elements of the Earth system will affect political support for environmental action. In our model, political support is one of the major determinants

of the work available for environmental action. This support - or 'political will' - is the result of the complex dynamics inherent in political, social, and economic systems. Varying political support will result in changes over a wide range of scales: from simple regulatory policies to deeper shifts in mindsets and paradigms in policymaking, all of which can unlock greater or lesser work on mitigation, adaptation, and restoration (Chan et al., 2020). The direct and indirect effects of destabilisation of elements of the Earth system could erode political

support for environmental action. For example, one response to significant environmental change can be increased migration (Parrish et al., 2020). This can increase socioeconomic inequality and conflict risk, factors that are known to drive authoritarian nationalism, which could, in turn, increase barriers to cooperative mitigation (Millward-Hopkins, 2022). Conversely, worsening direct effects, such as extreme weather events, could increase political will to act by serving as 'focusing events' for policymaking, increasing awareness of the

threat and spurring greater political activism that manifests in policy change for environmental action (Baccini and Leemann, 2021; Groff, 2021). The net effect of these connections is marked as ambiguous in figure 1, as represented by a question mark. This is due to a lack of literature exploring the overall effect of how the consequences of changes to the Earth system can erode and reinforce political support for mitigation, adaptation, and restoration. It is unclear whether the net effect of a more chaotic world will be to encourage far

more work done on environmental action or to crowd out this work as the focus shifts to, for example, disaster

response.

The third way that the work available to act on the transition varies arises from the processes of innovation that are encouraged by sustainability objectives. These partly occur as a consequence of development and innovation in economies, where penetration of technologies is accelerated by market dynamics, such as interactions
between research and development, learning by doing, economies of scale, and the spread of new social and market norms, all of which progressively reduce costs and increase acceptability (Smith, Stirling and Berkhout, 2005). This innovation is partly mediated by developments in politics and policymaking. For example, tax, subsidy, and regulation policies have been used in some countries to make electric vehicles cheaper and increase uptake (Sharpe and Lenton, 2021). In turn, these policy approaches can trigger 'positive' tipping points whereby
new technologies, societal norms, mindsets, and other innovations can rapidly out-compete incumbents (Systemiq, 2023). Crossing such a tipping point creates reinforcing feedback loops that accelerate uptake of the new approach or technology and that weaken resistance to change and support for incumbents. In the case of electric vehicles, reaching cost parity without tax or subsidy support can trigger reinforcing feedback loops of increasing returns to scale, with costs falling as production rises, increasing consumer demand for cheaper
alternatives, which also increase manufacturing and investment (Sharpe and Lenton, 2021). Such feedback mechanisms and the potential for positive tipping points exist across technological and energy systems, political mobilisation, financial markets, and sociocultural norms and behaviours, among other areas (Winkelmann et al., 2022). Consequently, the rate of change of the transition may be surprisingly large as it exceeds the expected capacity of social, economic, and political to undergo transformations. Therefore, in figure 1, the polarity
between pro-sustainability and political support for environmental action is positive in both directions: more support drives more innovation and more innovation drives more support, in a reinforcing feedback loop. In turn, innovation is marked as increasing work available.

Fourthly, the amount of work available for mitigation, adaptation, and restoration can be impacted by the effects of the transition itself. This is typically called transition risk in the context of risks to economic performance
because of pro-environment policies and action (FSOC 2021; TFCD 2021). We interpret transition risks as dynamics that can directly act to either increase or decrease the work available for further environmental action. For example, rapid changes in climate policy provide opportunities for renewable energy incumbents, reinforcing mitigation action (Mealy and Teytelboym, 2022). But this rapid action could also have the effect of disrupting financial stability, leading to credit rationing and falls in confidence and consumption (Semieniuk et
al., 2020). The spill over impacts could generally curtail investment, including in mitigation action. Yet we cannot find sufficient evidence in the literature to make a judgement on the net direct effect of transition risk on work expended on environmental action. Therefore, we have marked the polarity as ambiguous. There is also an indirect effect: transition risk can impact work available for environmental action through its impact on political will. For example, acceptance of transitions in energy systems is related to perceptions of distributive and
procedural justice (Evensen et al., 2018). Changes resulting from mitigation action that are perceived as unjust might curtail support, slowing the pace of decarbonisation. Conversely, as the transition happens to people and sectors, greater understanding and experience of the co-benefits of environmental action can reinforce support for further action (Cohen et al., 2021). However - again - we cannot find sufficient evidence in the literature to make a judgement on the net indirect effect of transition risk, via impacts on political will, and so on work

expended on environmental action. Therefore, we have marked the polarity as ambiguous. Finally, pro-sustainability innovation has an impact on transition risk. This includes, for example, how perceptions of climate change shift behavioural norms, which, in turn, impacts reputational risk for a given firm, economic activity, or sector (BIS, 2021). This dynamic could act to increase transition risk for those who are perceived as out of touch with shifting norms or decrease it for those who suddenly seem more 'in touch' with the times. A lack of evidence in the literature on the overall impact of transition risks and their interaction with innovation means this connection is ambiguous and is therefore illustrated with a question mark in figure 1.

**2.2 Derailment risk grows when work done is not sufficiently increasing**

The dynamics in figure 1 can be used to explore the overall impact on environmental action from the destabilising consequences of changes in the Earth system. We can identify two broad and opposing illustrative scenarios.

In the first, the reaction to the direct and indirect consequences of changes in the Earth system act to increase work expended on environmental action and reinforce political support (and so all respective connections are positive polarities in figure 1). This is partly driven by higher levels of adaptation: societies are better able to handle worsening direct and indirect impacts and so more work is available for environmental action and less political attention is sapped by crisis response. Transition risks have a net effect of increasing opportunities for environmental action as firms and sectors respond proactively to shifts in norms and policymaking. Pro-sustainability innovation and political support create a reinforcing feedback of societal and economic change that accelerates the transition. Therefore, derailment risk has been kept in check: work done on environmental action has increased even as societies have become more stressed by the consequences of changes in the Earth system. This scenario sees cascading and reinforcing positive tipping points that enable societies to achieve Earth system stewardship sufficient to avoid crossing a planetary threshold of cascading biophysical feedbacks and tipping points (Steffen et al, 2018; Lenton et al., 2022).

In the second scenario, the reactions to the direct and indirect consequences of changes in the Earth system act to limit or even reduce the amount of work done. Adaptation is insufficient to protect socio-economic systems from escalating impacts, which sap ever greater attention and resources from environmental action, and cause wider destabilisation that erodes political support. Transition risks - such as financial instability from rapid mitigation responses impacting investment decisions (Battiston et al., 2017) - have a net effect of further eroding work done. This is derailment risk in full effect. An overall reinforcing feedback loop is created in which the destabilising 'symptoms' of changes in the Earth system increasingly erode work done on tackling root causes (IPPR, 2023). Crucially, derailment risk increases if work done is not increasing sufficiently to outpace the effects emerging as a consequence of system feedback loops. This is because growing Earth system feedback loops have two effects on the work done. Firstly, they increase the amount of work that is needed on the Earth system; for example, climate feedbacks, including forest dieback, wildfires, and permafrost thaw, can increase sources and decrease sinks of greenhouse gas emissions, driving more warming (Ripple et al., 2023). Secondly, these feedbacks create more severe direct and indirect effects on societies resulting from changes in the Earth system. In this way, we can see derailment risk as representing a set of socio-economic feedbacks that interact with the set of intrinsic biophysical feedbacks identified in Steffen et al., 2018 that could push the Earth system

over a hypothesised planetary threshold beyond which spiralling requirements on work are needed to arrest an accelerating descent into a catastrophic 'Hothouse Earth' state.

## 3 Case study: AMOC collapse

To illustrate our theoretical framework, we explore a stylised scenario in which the activation of a climate tipping element - the collapse of the Atlantic Meridional Overturning Circulation (AMOC) - impacts work expended on environmental action, creating significant derailment risk. This scenario is illustrated in figure 2. The AMOC is an oceanic current system in the Atlantic Ocean driven by temperature and salinity differences

that brings heat from the southern hemisphere to northern latitudes. It is an important component of the regional and global climate system. Changes to ocean temperatures and salinity, themselves caused by climate change, can slow down the AMOC and could trigger its collapse (Lenton et al., 2008).

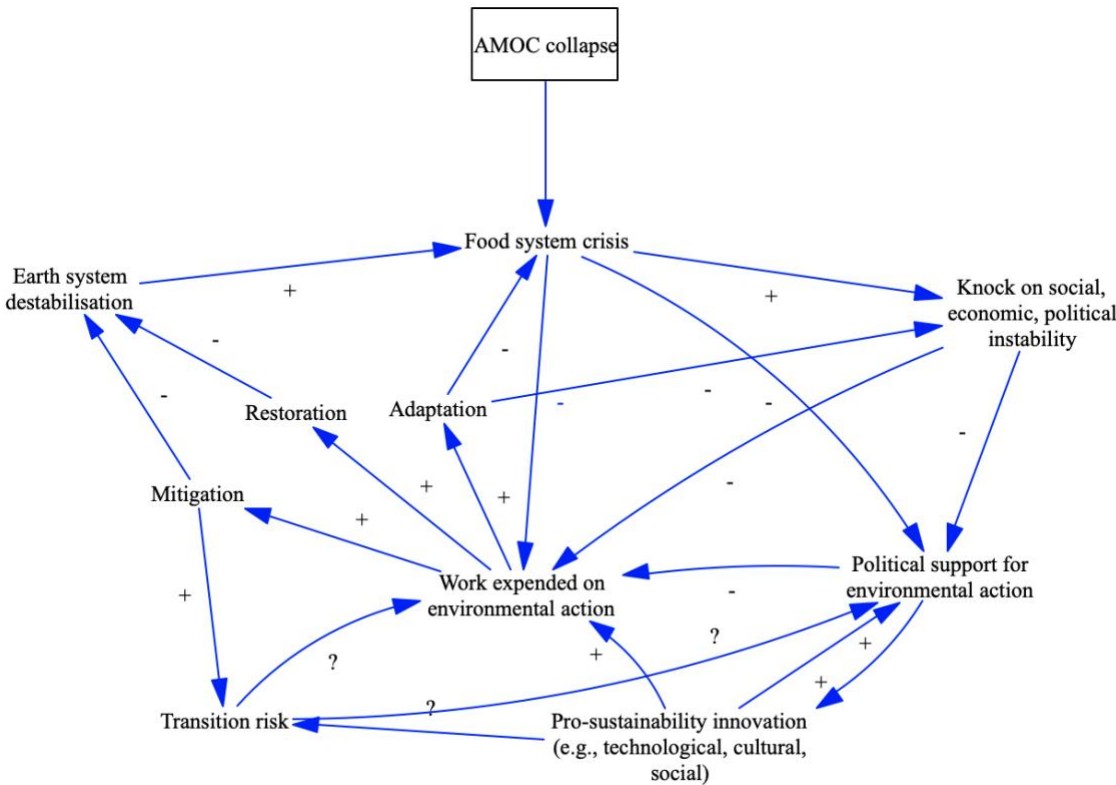

**Figure 2: Illustration of derailment risk affecting work expended on tackling climate change resulting**
**from the stylised scenario of the collapse of the Atlantic Meridional Overturning Circulation (AMOC).**

Collapse of the AMOC is an example of a subsystem of the Earth's climate system - called 'tipping elements' - that could pass a tipping point this century as a result of climate change (Armstrong McKay et al., 2022). Tipping points are, according to the Intergovernmental Panel on Climate Change (IPCC), irreversible levels of "change in system properties beyond which a system reorganises, often in a non-linear manner, and does not

return to the initial state even if the drivers of the change are abated. For the climate system, the term refers to a critical threshold at which global or regional climate changes from one stable state to another stable state"

(Babiker et al., 2018). Other examples include shrinkage of ice sheets, dieback of the Amazon rainforest, and disruption of monsoon systems (Armstrong McKay et al., 2022).

Evidence shows that the AMOC has weakened by around 15% since the mid-twentieth century (Caesar et al., 2018). The IPCC has concluded that there is only "medium confidence that the [AMOC] will not experience an abrupt collapse before 2100" and that the probability increases with higher global warming levels (Arias et al., 2021). While the latest climate models show large uncertainties in the assessment of a future collapse (Gong et al, 2022), models are known to overestimate the stability of the AMOC (Hofmann and Rahmstorf, 2009; Liu et al, 2017). One study predicts an AMOC collapse as soon as mid-century, with the period 2025 to 2095 as a 95% confidence range (Ditlevsen and Ditlevsen, 2023). If it were to occur, AMOC collapse would lead to large-scale impacts on the climate globally (Jackson et al., 2015).

The effects of these shifts are explored in more detail in OECD, 2021, which considers the possibility of AMOC collapse without underlying warming and at 2.5°C above the pre-industrial level as a significant risk befitting an assessment. The induced shift in climatic conditions of an AMOC collapse in either scenario would have profound impacts on agriculture across the world, posing a critical threat to food security globally (OECD, 2022). An AMOC collapse occurring alongside warming would substantially reduce the growing suitability of three major staple crops - wheat, maize, and rise - which provide the majority of global calories (OECD, 2021). Without underlying warming, nearly a quarter of the current area for wheat is lost, with a 16% loss for maize and a smaller change for rice (ibid). With 2.5°C of warming, approximately half of the remaining suitable land for wheat and for maize is lost, while there is a small increase in suitable area for rice (ibid). The authors of the OECD study concluded that "AMOC collapse would clearly pose a critical challenge to food security. Such a collapse combined with [temperature rises of 2.5°C above the pre-industrial level] would have a catastrophic impact" (ibid).

We have selected AMOC collapse for our stylised scenario because of the severity of its potential impacts and their global spread, and that the collapse cannot be ruled out this century. Additional effects of AMOC collapse include disruptions to monsoon systems, with all the knock-on effects for crop production, economic stability, and health and wellbeing these could bring, and dieback destruction of boreal forests in northern Europe and Asia, which constitutes a cascading impact on other parts of the climate system (OECD, 2021). These along with the severe direct impacts on food security explored above would have considerable and far-ranging indirect consequences, including on economic, social, and political stability (OECD, 2022).

Therefore, in our stylised scenario, we posit that the food system crisis caused by AMOC collapse acts to erode environmental action by redirecting work into emergency response to protect populations from food insecurity and to handle the wider destabilising consequences, which demand significant resources, labour time, and energy requirements that could otherwise be employed for environmental action. This also occurs indirectly as political support for environmental action is crowded out by the imperative of emergency response on a global scale in response to the catastrophic impacts to the food system and their cascading consequences. While there is evidence that the experience of natural disasters provokes a surge in pro-climate voting and politics, it is inversely possible that as food shocks become more frequent, this could reduce the adaptive capacity of society to respond to future shocks and thereby further crowd out capacity for work on environmental action (Baccini

and Leemann, 2021; Mehrabi 2020). With regards to transition risks, it is possible that an abrupt 'overcorrection' in climate policy stimulated in reaction to the effects of an AMOC collapse could itself have adverse systemic consequences, particularly for the financial system, which would also crowd out capacity for work on environmental action (Battison et al., 2017). Even if the reinforcing feedback loops between pro-sustainability innovations and political support continues, their effect might be insufficient to compensate for

the direct and indirect reductions of work done on mitigation, adaptation, and restoration.

Overall, in this scenario, the world is pushed into a state of spiralling derailment risk. The resultant reduction in work expended on re-stabilising elements of the Earth system combined with the impact on biophysical feedbacks from AMOC collapse would cause escalating direct and indirect impacts, further exacerbating derailment risk. We posit that this dynamic could make it progressively more difficult to rally political support

and expend the work needed to establish a trajectory in which humanity remains within a safe and operating space. Instead, interacting socioeconomic and biophysical feedbacks could create a cascade of direct and indirect impacts with multiple planetary boundaries being breached. One end point for such reinforcing feedbacks could be continued warming of the climate putting the Earth on a course towards the hypothesised 'Hothouse' state.

**4 Implications for policy strategies**

In essence, derailment threatens our collective agency to correct changes to the Earth system. In the extreme, derailment risk could fatally constrain this agency, as the illustrative scenario of an AMOC collapse explores. Therefore, derailment risk has profound implications for policy strategies.

One area is in relation to scenarios. Five 'Shared Socioeconomic Pathways' (SSPs) now serve as some of the

main scenarios exploring interactions between human societies and the natural environment over the 21st century (O'Neill et al., 2017). As a result, they are a major guide to policy responses. The SSPs consider projected global socioeconomic changes up to 2100 - including population, urbanisation, and GDP - and the subsequent challenges to mitigation and adaptation, enabling an integrated analysis of many factors determining climate action (Riahi et al., 2017). For example, SSP1 ("Sustainability: Taking the green road") sees rapid

technological change, more globally equal development, and a greater focus on environmental sustainability, all resulting in low challenges to mitigation and adaptation. In contrast, SSP4 ("Inequality: A road divided") has low challenges to mitigation resulting from high but unequally distributed technological development and large challenges to adaptation due to inequality and persistent poverty in some parts of the world.

However, the SSPs do not directly consider the connection between the consequences of changes in the Earth's

climate and work available for re-stabilisation of stressed biophysical systems. A major consequence of our model of derailment risk is that this omission could be a dangerous blind spot in how the SSPs are guiding policymaking on re-stabilisation. It cannot be assumed that collective work on the Earth system - and societies' ability to muster growing amounts of work - will inevitably grow, both directly, through more technological capacity and resources, and indirectly through more political will. This is the case whether or not it is assumed

that continued growth in material production and consumption is compatible with planetary boundaries.

Overall, the work done to re-stabilise elements of the Earth system in order to avoid passing a planetary threshold will be impacted by a more complex set of feedbacks than are considered in the SSPs. A failure to capture these feedbacks can lead to a significant underestimation of the societal risks of changes to the Earth system and a misinterpretation of the collective ability to recover a safe operating space and the simultaneous ability to effectively manage the consequences of exceeding it. This in addition to the exclusion in the SSPs of a wider set of interactions between climate change and other areas of Earth system change, such as feedbacks between rising temperatures and biogeochemical flows. These 'missing' interactions may be very important with regards to destabilising socio-ecological system feedback loops. It is imperative that these feedbacks are included in the climate change mitigation scenarios. Failure to capture interactions between human societies and the natural environment means vital derailment risk dynamics will continue to be omitted from policy-relevant scenarios.

Beyond applications for scenarios, our model is an attempt to identify areas for the mitigation of derailment risk. These correspond to the connections on the systems diagrams in figures 1 and 2. A primary means to respond to derailment risk is to increase work done on re-stabilising elements of the Earth system to attenuate conditions from which the risk arises. This can be driven by greater political support for action and the interaction between innovations in, say, social and political movements which can drive this support, or using policies that target rapid changes in the rollout of clean technologies and behaviours (Winkelmann et al., 2022). A large range of these positive social and economic tipping points have now been identified (Sharpe and Lenton 2021; Systemiq, 2021; Systemiq, 2023).

However, the processes by which these positive tipping points can occur will have to be made robust to derailment risk. For example, the severe impacts on food security considered in the AMOC case study might create chaotic conditions that crowd out political support for policies that drive positive tipping points. In response, the processes by which positive tipping points are triggered should be made more robust in withstanding the direct and indirect impacts of the destabilisation of Earth system elements. This can be done directly through adaptation that reduces the effects of these impacts on work available for environmental action. It can also occur indirectly, by ensuring the drivers of political support for environmental action and that drive innovations that trigger positive tipping points are made more resilient. In this regard, we should see adaptation as an enabler of mitigation under conditions of escalating destabilisation within the Earth system: the sustainability transition itself needs to be made more resilient.

In this way, derailment risk bolsters the case for 'transformational adaptation'. These are adaptations that fundamentally change the characteristics of human and natural systems so that their capacity to cope with hazards is increased (Pörtner et al., 2022). This is in distinction to 'incremental adaptation', which refers to adaptations to specific system components to protect against given climate risks, such as modifying infrastructure to handle sea level rise (Kates, Travis and Wilbanks, 2012). Because of the connection between direct and indirect consequences of changes in the Earth system and political support, concepts of resilience in this regard need to extend to concepts of justice, fairness, trust, and participation, all of which are factors impacting acceptance of the transition (Gölz and Wedderhoff, 2018; Mundaca et al., 2018; Evensen et al., 2018), ensuring political support is maintained and deepened even as incentives for protectionism and competition might grow. Resilience also extends to psychological and emotional factors. Studies of anxiety over

climate change report that these feelings have negatively impacted day-to-day functioning and that anxiety and distress are correlated with perceptions of inadequacy and betrayal on the part of governments and leaders (Hickman et al., 2021). Being wise to the emotional and psychological consequences of escalating impacts is an important factor affecting both perceptions and action (Brosch, 2021).

**5 Conclusion**

There is now a considerable and growing body of research that explores the risks to societies that arise from changes in the Earth system, and the transition risks that result from actions to re-stabilise stressed biophysical systems. However, there is limited exploration of how the effects of changes in the Earth system will present challenges to societies' ability to undertake the work necessary to redress those changes. This is a dangerous gap. In this paper we have introduced a conceptual socio-ecological systems model that explores this area,

applying it to a scenario of the activation of a climate tipping element. This serves as a case study in which the escalating consequences of the tipping event divert work and political support away from environmental action, thus amplifying destabilisation within the Earth system.  This further increases the chance of passing a planetary threshold over which human agency to re-stabilise the natural world is severely impaired. We present this scenario as an example of the risk that the sustainability transition could be increasingly undermined by the

worsening impacts of climate and ecological change. We call this derailment risk.

Our model provides a simple qualitative mapping of the dynamics of this risk by identifying potential feedback loops. Further work in this area is urgently needed. This should build on emerging methods for understanding and mapping cascading and systemic risks within socioeconomic systems resulting from changes in the Earth system (see, for example, UNDRR, 2021) and similar areas of study and early warning systems for feedbacks

and non-linear dynamics in the Earth system, including tipping points (Bury et al., 2021). Within this, a natural extension of our qualitative model is through translation into a dynamical system with quantitative values drawing on the latest understanding of changes in Earth system elements, social and economic systems, and their interactions. Relevant examples of similar exercises include Lade et al., 2019. Furthermore, mapping of derailment risk should pay particular attention to concepts of fairness and equality, which are currently assumed

within the political dynamics within our simple model but are crucial factors determining cooperation on environmental action.

Our analysis leads us to conclude that it is essential that sustainability transition policies are designed to withstand large-scale turbulence in biophysical and socioeconomic systems. Optimal strategies are not necessarily the most resilient. Failure to provide sufficient work for the sustainability transition risks its partial

or even complete derailment. Both protecting and enhancing our collective ability to trigger positive social and economic tipping points, even as chaotic events grow, could create a more powerful counter-effect to the increasingly dire effects of changes in the Earth system, avoiding spiralling disaster. In developing a simple qualitative model, this article hopes to be of value to the research community, policy makers, and wider society in driving understanding and action on derailment risk. This can help avert outcomes that see societies tip into

an almost unimaginably more turbulent and dangerous world.

**Code and data availability**

The model was developed using Vensim. Original files can be made available on request.

**Author contributions**

LL developed the model, JD advised on development. All authors drafted the manuscript.

**Competing interests**

At least one of the (co-)authors is a member of the editorial board of Earth System Dynamics.

**Financial support**

None of the authors had any financial support in the development and writing of this manuscript.

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
