# Peer review of "Derailment risk: A systems analysis that identifies risks which could derail the sustainability transition"

_EGUsphere, 2023_

## Author Response (AR1)

**Derailment risk: A systems analysis that identifies risks which could derail the sustainability transition**

*AUTHORS' RESPONSE DOCUMENT*

**In response to comments from reviewer 1**
*This manuscript develops a conceptual socioecological model to illustrate derailment risk of a sustainability transition. This is an important topic in furthering our understanding of the complex interactions between the Earth system and human systems. The manuscript is well written. I have several remarks which should be addressed before the manuscript can be considered for publication.*

We thank the reviewer for their careful review and these positive comments. Below we respond to specific comments and detail the responses and improvements that we incorporated into a revised manuscript.

*I have problems with the term "destabilization". For me this term implies that the Earth system becomes unstable, which in term implies either a hothouse runaway climate or much more variability. You might just refer to a shift to another climate state (which might then be stable again). It might be best if the term is defined.*

We appreciate that the term "destabilization" can imply, for example, trajectories toward a hypothesised hothouse runaway climate. This was not our intention. Instead, it was to highlight that it is not 'just' the climate system that is experiencing change but also other elements of the Earth system. We do, though, want to make some link to the potential for derailment risk, as we define it, to contribute toward a reduction in work which could increase the potential for a transition to a new, more dangerous state. We agree that we need to carefully specify our terminology here. The destabilisation we refer to is biophysical and socio-economic. Our central thesis is that interactions between biophysical and socio-economic systems can produce feedback loop dynamics that have reinforcing effects. Such effects could significantly degrade human societies' ability to effectively respond to the challenges of climate and ecological change.

Therefore, we have referred to 'changes in the Earth system' and 'destabilisation of biophysical systems' to ensure differentiation between these changes and any overall variability or destabilisation in the Earth system.

*Lines 230-235: Here you mentioned that agricultural area might be lost in the future. Can you please provide references for this.*

The loss of agricultural area is explored in OECD, 2021, but has not been referenced, apologies. This was corrected.

*Line 235: "Such a collapse combined with climate change". Isn't the collapse due to climate change?*

The quote is taken from the OECD paper (OECD, 2021). It is in reference to modelling of an AMOC collapse that explores the effect of the collapse on agriculture at current levels of warming, and then at 2.5C of warming. The authors are comparing the two scenarios. We have made this clearer in the revised manuscript.

*Line 239: What do you mean by work? Resources/funding/effort or actual labour work?*

Our concept of work in relation to derailment risk is intentionally broad, encompassing physical work, resource use, funding, all the way to less tangible factors, such as political support, which determines other forms of work. This is developed in section 2. We have added more clarity to our definition.

*Line 264: "... in the round". What do you mean by that expression?*

In this context, we meant "in the round" to mean that SSPs do not include a wider set of interactions between climate change and other areas of Earth system change, such as feedbacks between rising temperatures and biogeochemical flows, which present risks for societies and economic systems. These 'missing' interactions may be very important with regards emergent destabilising socio-ecological system feedback loops. This has been more carefully explained in the text.

*Line 281: Can you make your model more quantitative? ESD aims to publish quantitative studies. One way would be to make a systematic literature study based on studies of each link of your model.*

*Also links with a "?": are previous studies inconclusive or are there no studies at all on those links? How robust are the "+" and "-" links?*

Our objective in this manuscript is to produce a qualitative model output that would be of value to policy makers and wider society. In this manuscript we first need to define the scope of the modelling activity and identify what we assess to be first-order terms and dynamics. In doing so we can establish the concept of 'derailment risk' and thus a qualitative model is the best initial step for doing so. In the revised manuscript we have provided more detailed supporting literature with regards the interactions and polarities of such interactions. We have also acknowledged where there is insufficient literature toreach judgements on the polarities (hence marking them with a "?"). Additionally, it is worth noting that this is partly the result of 'known unknowns' whereby it might be reasonable to assume that interactions exist, but it is not possible to determine polarity. We have also provided a discussion of more quantitative extensions to this concept and modelling.

**In response to comments from reviewer 2**

*The paper "Derailment risk: A systems analysis that identifies risks which would derail the sustainability transition" by Laurie Laybourn, Joseph Evans and James Dyke nicely illustrates the risk emerging to a sustainability rather than earlier works that elaborate on physical risks and transition risks. Methodologically, the paper develops a feedback diagram between Earth system destabilisation, Earth system impacts, political support and transition risks towards the ability to tackle the root causes of Earth system destabilisation. Overall, I think that the paper tackles an important issue whether societies are (and under which circumstances) able to work on environmental action. The authors apply their framework then on a climate tipping point, the Atlantic Meridional Overturning Circulation (AMOC). I think the paper is well written but should take into account the following comments before publication:*

We thank the reviewer for their careful review and the very useful opening summary of our manuscript. Below we respond to specific comments and detail the responses and improvements that were incorporated into the revised manuscript.

*The authors explain well in which contexts derailment risks are relevant in case of negative feedbacks between political instability to political support to work expended. However, I wonder whether a different case study than the AMOC might be easier to put into the context of the feedback diagram that the authors develop (e.g. deforestation of the Amazon rainforest?). If the authors decide to stick with the AMOC example (which I am happy to support), I would recommend:*

- *Can the authors set the current state of research on the AMOC-tipping better into context? so where do we stand with respect to a potential AMOC tipping (some helpful references might be:*
    - *Caesar, L., Rahmstorf, S., Robinson, A., Feulner, G. and Saba, V., 2018. Observed fingerprint of a weakening Atlantic Ocean overturning circulation. Nature, 556(7700), pp.191-196.*
    - *Ditlevsen, P. and Ditlevsen, S., 2023. Warning of a forthcoming collapse of the Atlantic meridional overturning circulation. Nature Communications, 14(1), p.4254.*
    - *Jackson, L.C., Kahana, R., Graham, T., Ringer, M.A., Woollings, T., Mecking, J.V. and Wood, R.A., 2015. Global and European climate impacts of a slowdown of the AMOC in a high resolution GCM. Climate dynamics, 45, pp.3299-3316.*

We are very grateful for these reference suggestions. We have reviewed these and updated the manuscript accordingly, setting the state of AMOC research in the wider context, using all of the references listed (and others). We have also provided additional explanation of why we have used the AMOC in the case study: we believe this example to be particularly clear in the risks presented by, for example, changes to agriculture and how these could lead to derailment risk.

*242-245: While an AMOC shutdown indeed changes monsoon patterns, it decreases temperatures regionally, and partially also in regions where Xu et al., 2020, PNAS assess the largest risk of leaving the human climate niche (e.g. Sahel, Arabian Peninsula), see e.g.*

*Jackson et al., 2015, Climate Dynamics. Therefore, I would conclude that AMOC impacts on the human climate niche are inconclusive to say the least. Therefore, I think that the argumentation in L. 242-245 should be sharpened.*

We thank the reviewer for this clarification. We have more carefully specified the impacts AMOC may have on social-economic systems and have removed reference to the 'climate niche', recognising how inconclusive these impacts are.

*Please check references carefully. I couldn't find the following references in the reference list: Arneth et al., 2019; Steffen et al., 2018; University of Exeter et al., 2023, …*

We thank the reviewer for catching this omission. Referencing mistakes have been addressed and the referencing overall has been checked.

*I like how the authors reason their feedback diagram in figure 1 and explain their links afterwards in the main text. However, I agree with the other reviewer that a systematic literature review would significantly strengthen the paper but at least a more thorough referencing in section 2 would be very helpful, I think.*

We have taken on board this and the related comment from Reviewer 1. In doing so, we have included a more systematic use of literature to substantiate the interactions we sketch out in our conceptual model and have given commentary where we the literature does not provide enough for us to ascertain a positive or negative polarity.

*Optional: I like how the authors put their conceptual framework into context in the discussion and conclusion. I wondered if the authors have any idea how it could be possible to translate the feedback diagram into a dynamical system, e.g. using differential equations. Maybe similar to work that has been done in this manuscript:*
*Lade, S.J., Norberg, J., Anderies, J.M., Beer, C., Cornell, S.E., Donges, J.F., Fetzer, I., Gasser, T., Richardson, K., Rockström, J. and Steffen, W., 2019. Potential feedbacks between loss of biosphere integrity and climate change. Global Sustainability, 2, p.e21.*

A quantitative model using dynamical systems is very much within the scope of our aspirations for this work. Given the complexity of the biophysical and socio-economic systems it can be very challenging to produce useful formalism that captures the various interactions. We hope that this study can firmly establish the boundaries of such quantitative modelling exercises. As such, we have provided more commentary on future work (including citing the study referenced, which we see as a strong example on which to draw when developing a more quantitative version of our model).